# Power Converter Resonant Control for an Unbalanced and Non-Constant Frequency Supply

**DOI:** 10.3390/s23104884

**Published:** 2023-05-19

**Authors:** Jaime Rohten, Felipe Villarroel, José Silva, Esteban Pulido, Fabián Pierart, Johan Guzmán, Luis García-Santander

**Affiliations:** 1Department of Electrical and Electronic Engineering, Universidad del Bío-Bío, Concepción 4030000, Chile; 2Department of Engineering Science, Universidad de Los Lagos, Puerto Montt 5480000, Chile; 3Department of Electrical Engineering, Universidad Técnica Federico Santa María, Valparaíso 2390136, Chile; 4Department of Mechanical Engineering, Universidad del Bío-Bío, Concepción 4030000, Chile; 5Department of Electrical Engineering, Universidad de Concepción, Concepción 4030000, Chile

**Keywords:** resonant control, microgrids, unbalanced three-phase system

## Abstract

Distorted voltage supplied as unbalanced and/or non-constant frequency can be found in weak grids, such as microgrids, or systems working in islanding mode. These kinds of systems are more sensitive under load changes. Particularly, an unbalanced voltage supply may be produced for large, single-phase loads. On the other hand, the connection/disconnection of high current loads may lead to important frequency variation, especially in weak grids where the short circuit current capacity is reduced. These conditions make the control of the power converter a more difficult task, because of the variations in the frequency and unbalancing. To address these issues, this paper proposes a resonant control algorithm to deal with variations in the voltage amplitude as well as grid frequency when a distorted power supply is considered. The frequency variation is an important drawback for resonant control because the resonance must be tuned at the grid frequency. This issue is overcome by using a variable sampling frequency in order to avoid re-tuning the controller parameters. On the other hand, under unbalanced conditions, the proposed method relaxes the phase with lower voltage amplitude by taking more power from the other phases in order to help the stability of the grid supply. To corroborate the mathematical analysis and the proposed control, a stability study is performed, including experimental and simulated results.

## 1. Introduction

Controlled power converters have grown in acceptance and are therefore widely used in many applications. To enumerate some of them, for instance, they are used in motor drives, renewable energy, and active filters, among others [1]. Additionally, many topologies have been developed to improve different aspects of these power converters [2,3]. The most common power converter controllers are linear, nonlinear, predictive, and resonant [4,5,6,7]. The controller is tied to the environment, the topology, and the requirements of the process involved. Particularly, this paper deals with a non-constant frequency environment, as well as an unbalanced voltage supply condition. These problems are of particular interest to weak grid sources, microgrids, and systems working in islanding mode [7,8,9].

On the one hand, frequency variations are related to differences in the power supplied and the power consumed. This usually has important effects on the frequency value, especially on weak grids, where the short-circuit capacity is notably reduced. This reduction has been documented as up to 100% of the frequency variation on some microgrids, such as ships and aircraft power systems [10]. On the other hand, unbalanced conditions may be related to single-phase loads connected to the aforementioned system, leading to the overload of one of the phases; because of the line impedance effect, the voltage is reduced in the overloaded phase at the point of coupling. Thus, the issues of frequency variations and unbalanced supply voltage are key problems to be overcome by the controller [11].

Proportional resonant control is an attractive option to be used for the control of active rectifier systems [12]. For instance, resonant control allows the achievement of a desired dynamic response without relying on reference frame transformations. The total algorithm is very small and easy to implement on digital boards with a reduced computation effort. However, the controller is sensitive to the tuned frequency; so, to apply this controller to a variable-frequency voltage supply, an adaptive algorithm to overcome the supply deviations is required.

Frequency variations affect controller tuning [5], which is usually set for a specific grid frequency. Under grid frequency changes, the controller should take this information into account. In order to overcome this issue, the sampling time is changed as a function of the grid frequency in order to keep a constant number *N* of samples per cycle. This approach allows to keep the controller’s parameters constant; i.e., the poles do not need to be changed as a function of the grid frequency as demonstrated in this paper. To accomplish *N* samples per second, a variable sample controller should be introduced [10] in the digital board. Due to the variable sample time algorithm’s quick dynamic response, it is possible to neglect this dynamic in the rectifier current and voltage controller design [13].

An unbalanced voltage supply also introduces an essential challenge for rectifier control [14,15]. The main problem is the current waveform imposed by the rectifier control. The current in each phase can be adjusted independent of the supplied voltage being balanced or unbalanced, depending on the voltage imbalances. Assuming that imbalances in a weak grid source are produced by single-phase overloads, to help the network stability, the control should demand less power from the overloaded phases. Thus, we propose to compute the current references as a function of the unbalanced voltage to reduce the power consumed by the overloaded phases.

To corroborate the designed controller, a stability analysis is required to ensure a stable closed-loop system. The results, including simulation and experimentation results, show the feasibility of the proposed controller in a wide frequency range and under unbalanced voltage conditions.

This paper is organized as follows: Section 2 introduces the model of the proposed converter system with variable grid frequency, which was used in the controller design. Section 3 presents the proposed resonant-control-based scheme together with the current reference generation required to operate under an unbalanced condition. Section 4 studies the evolution of the system’s poles locations when the grid frequency changes. Section 5 presents the simulation and experimental validation of the proposed method. Finally, the main conclusions of this study are presented in Section 6.

## 2. Power Converter Model

The topology in Figure 1 shows a rectifier connected to a distorted grid source that supplies a controlled rectifier to fix the voltage and frequency in a desired value independent of the grid quantities’ variations. The main objective is to control voltage amplitude and frequency using the rectifier to maintain a desired DC link voltage. When the DC link voltage is constant and close to a reference, the inverter can be easily controlled. Thus, the rectifier’s control is significantly more challenging, as it is connected to a distorted environment.

The system behavior can be modeled through Kirchhoff’s laws. Writing the equations leads to the controller design. The rectifier input current depends on the grid and rectifier voltages and can be written as:(1)$${v_{g}}^{abc}=L_{g}\frac{d{i_{g}}^{abc}}{dt}+R_{g}{i_{g}}^{abc}+{v_{\mathrm{go}}}^{abc},$$
where ${v_{g}}^{abc}$ is the grid voltage vector, $L_{g}$ is the filter inductor, $R_{g}$ is the parasitic resistor, and ${v_{\mathrm{go}}}^{abc}$ is the rectifier input voltage. The DC voltage model, which is used to link the rectifier and the inverter, can be represented as:(2)$$C_{dc}\frac{dv_{dc}}{dt}=i_{g}^{dc}-i_{L}^{dc},$$
where $C_{dc}$ is the DC link capacitance value, $v_{dc}$ the dc link voltage, $i_{g}^{dc}$ is the rectifier output current, and $i_{L}^{dc}$ is the inverter input current.

For SPWM modulation, a comparison between a carrier with a modulating signal results in the switching states [16]; thus, the average value of the voltage injected by the converter ${v_{\mathrm{go}}}^{abc}$ can be expressed as:(3)$${v_{\mathrm{go}}}^{abc}=G_{ac}{m_{g}}^{abc}v_{dc},$$
and the DC current is expressed as:(4)$$i_{g}^{dc}={\left(G_{ac}{m_{g}}^{abc}\right)}^{T}{i_{g}}^{abc},$$
where $G_{ac}$ is the modulation technique gain, and $m_{g}$ are the modulating signals of the grid-side converter.

On the other hand, the equation that describes the inverter model is defined as:(5)$${v_{\mathrm{Lo}}}^{abc}=R_{Lf}{i_{\mathrm{Lo}}}^{abc}+L_{Lf}\frac{d{i_{\mathrm{Lo}}}^{abc}}{dt}+{v_{L}}^{abc},$$
(6)$${i_{\mathrm{Lo}}}^{abc}=C_{Lf}\frac{v_{L}^{abc}}{dt}+{i_{L}}^{abc},$$
where:(7)$${v_{\mathrm{Lo}}}^{abc}=G_{ac}{m_{L}}^{abc}v_{dc},$$
and
(8)$$i_{L}^{dc}=G_{ac}{\left({m_{L}}^{abc}\right)}^{T}{i_{\mathrm{Lo}}}^{abc},$$
where ${m_{L}}^{abc}$ are the modulating signals of the load-side converter.

The system in Figure 1 allows decoupling the load with respect to the distorted grid source, with the load frequency and amplitude determined by the load-side inverter control algorithm; the literature shows several ways to control them, such as those in [16,17,18,19,20]. Therefore, this study did not focus on load inverter control but only on the grid-side converter, which must deal with a distorted source (as the one presented in microgrids and weak grid systems), delivering, despite the unpredictable behavior of the grid, a controlled DC voltage to feed the load-side inverter with the desired DC link voltage.

## 3. Power Converter Resonant Control

This work focused on the rectifier’s controller when connected to a distorted supply voltage, considering frequency and amplitude variations together with unbalanced conditions. Herein, the development of the strategy and its operation are comprehensively detailed, emphasizing the operation under the aforementioned conditions.

### 3.1. Control of the DC Link Voltage

To provide the power demanded by the load (inverter), the rectifier drains power from the grid supply to the DC link. The power consumed by the rectifier can be separated for control purposes into (*i*) active power to charge/discharge the DC link capacitor, and (*ii*) reactive power to inject a desired reactive power to the grid. Therefore, the DC link voltage control imposes an active power that, avoiding the RL input filter, can be defined as:(9)$$p_{g}^{\mathrm{ref}}=3v_{gp}i_{gp}^{\mathrm{ref}},$$
where the current reference $i_{gp}^{\mathrm{ref}}$ is imposed to track the DC link voltage reference by an internal current control loop. The reference $i_{gp}^{\mathrm{ref}}$ can be readily found from (Equation 9), leading to:(10)$$i_{gp}^{\mathrm{ref}}=p_{g}^{\mathrm{ref}}/\left(3v_{gp}\right).$$

The power reference $p_{g}^{\mathrm{ref}}$ required to maintain the DC link voltage tracking its own reference is generated by a discrete-time PI controller with the following transfer function:(11)$$h_{c}^{vdc}\left(z\right)=\frac{k_{1}+k_{2}z^{-1}}{1-z^{-1}},$$
where $k_{1}=k_{c}\left(1+\frac{T_{s}}{2T_{i}}\right)$ and $k_{2}=k_{c}\left(-1+\frac{T_{s}}{2T_{i}}\right)$, where $k_{c}$ and $T_{i}$ are the gain and integrative time of the PI controller, respectively. Thus, (Equation 11) can be written as a difference equation for its discrete-time implementation as:(12)$$p^{\mathrm{ref}}\left(k\right)=p^{\mathrm{ref}}\left(k-1\right)+k_{1}e_{vdc}\left(k\right)+k_{2}e_{vdc}\left(k-1\right),$$
where $e_{vdc}$ is the voltage error $e_{vdc}={\left(v_{dc}^{ref}\right)}^{2}-{\left(v_{dc}\right)}^{2}$. Finally, difference Equation (Equation 12) together with (Equation 10) are simultaneously used to regulate the DC link voltage. Thus, the power required from the grid is given by:(13)$$p_{g}^{\mathrm{ref}}\left(k\right)=p^{\mathrm{ref}}\left(k\right)+v_{dc}i_{L}^{dc},$$
neglecting the load losses at the input filter.

The system diagram is shown in Figure 2, where the DC link voltage is an integrator, as illustrated in [5].

### 3.2. Rectifier Power Factor Control

When reactive power injection is desirable, this component must be included in the current control loop. Then, the two components of the power are separated in the complex plane as:(14)$${\vec{s_{g}}}^{\mathrm{ref}}=p_{g}^{\mathrm{ref}}+jq_{g}^{\mathrm{ref}},$$
where ${\vec{s_{g}}}^{\mathrm{ref}}$ is the apparent power, and $p_{g}^{\mathrm{ref}}$ and $q_{g}^{\mathrm{ref}}$ are the active and reactive power components, respectively.

On the other hand, the power factor is the proportion of the active power with respect to the apparent power, which is represented by:(15)$${\left(pf\right)}^{\mathrm{ref}}=p_{g}^{\mathrm{ref}}/\left|{\vec{s_{g}}}^{\mathrm{ref}}\right|=p^{\mathrm{ref}}/\sqrt{{\left(p_{g}^{\mathrm{ref}}\right)}^{2}+{\left(q_{g}^{\mathrm{ref}}\right)}^{2}}.$$

The reactive power reference can be found from (Equation 15) as:(16)$$q_{g}^{\mathrm{ref}}\left(k\right)=\pm p_{g}^{\mathrm{ref}}\left(k\right)\sqrt{1/{\left({\left(pf\right)}^{\mathrm{ref}}\right)}^{2}-1}=\pm k_{pf}p_{g}^{\mathrm{ref}}\left(k\right),$$
where a positive or negative value represents operation with inductive or capacitive power factor and is decided by the user. $k_{pf}=0$ represents a unitary power factor.

### 3.3. Voltage Amplitude Estimator

The grid supply voltage can be unbalanced, which may indicate an overloaded phase. To further use this information for the current control loop, a voltage amplitude estimation algorithm is introduced in order to know the voltage amplitude of each phase.

The grid voltage vector ${v_{g}}^{abc}$ can be expressed as:(17)$$v_{g}^{abc}={\left[V_{g}^{a}\sin\left(\theta_{g}\right)\;V_{g}^{b}\sin\left(\theta_{g}-\frac{2\pi }{3}\right)\;V_{g}^{c}\sin\left(\theta_{g}+\frac{2\pi }{3}\right)\right]}^{T},$$
where $V_{g}^{a}$, $V_{g}^{b}$, and $V_{g}^{c}$ are the voltage amplitudes of its separate phases, and $\theta_{g}$ is the instantaneous voltage vector position.

The amplitude can be found from the rms value of the instantaneous voltage waveform, which is computed as:(18)$${\left(V^{l}\right)}^{2}\left(k\right)=\frac{2}{N}\sum_{i=k-N+1}^{k}{\left(v_{g}^{l}\right)}^{2}\left(k\right),$$
with *l* representing the phase *a*, *b*, or *c*, leading to:(19)$${\left(V_{g}^{l}\right)}^{2}\left(k\right)={\left(V_{g}^{l}\right)}^{2}\left(k-1\right)+\frac{2}{N}\left[{\left(v_{g}^{l}\right)}^{2}\left(k\right)-{\left(v_{g}^{l}\right)}^{2}\left(k-N\right)\right],$$
where $V_{g}^{l}$ is the voltage amplitude for the phase *l* at time *k*, and *N* is the number of samples per period imposed by the controller as is defined in the following. The diagram for this purpose is shown in Figure 3.

### 3.4. Current References

The current references follow the require active and reactive power from (Equation 15) and (Equation 16) but consider the voltage amplitude estimated as illustrated in (Equation 19). Thus, the current related to the phase with reduced amplitude is also reduced, and the current associated with the higher-amplitude voltage also has a higher amplitude. The aforementioned requires the identification of the higher-voltage amplitude (among each phase) and normalize the rest as a function of the higher voltage amplitude. In this section, the algorithm presents how the amplitude current reference is set for each phase, including the phase shift associated with the power factor.

When the unbalanced condition is produced by overload in some phase (or phases) of the grid supply, the overloaded phase (or phases) may be mitigated by decreasing the current load on it (them). To this end, the voltage amplitude (Equation 19) of each phase can be computed and then compared with each other, seeking imbalances. Therefore, the maximum amplitude among the phases is:(20)$${\left[V_{gmax}\left(k\right)\right]}^{2}=\max_{l=a,b,c}{\left[V_{g}^{l}\left(k\right)\right]}^{2};$$
thus, the proportion of every other voltage with respect to the maximum value $V_{gmax}^{l}$ can be written as:(21)$${\left[V_{gmax}^{l}\left(k\right)\right]}^{2}={\left[V_{g}^{l}\left(k\right)\right]}^{2}/{\left[V_{gmax}\left(k\right)\right]}^{2},$$
with l={a,b,c}.

The abc current references are imposed by the active and reactive power references (Equation 13) and (Equation 16), which means a current in phase with a supply voltage (active power) and π/2 radians (reactive power) phase-shifted with respect to the supply voltage. Assuming the voltage in (Equation 17), the current references have the following form:(22)$${\left(i_{g}^{a}\right)}^{\mathrm{ref}}=I_{gp}^{a}\sin\left(\theta_{g}\right)+I_{gq}^{a}\cos\left(\theta_{g}\right),$$
(23)$${\left(i_{g}^{b}\right)}^{\mathrm{ref}}=I_{gp}^{b}\sin\left(\theta_{g}-\frac{2\pi }{3}\right)+I_{gq}^{b}\cos\left(\theta_{g}-\frac{2\pi }{3}\right),$$
(24)$${\left(i_{g}^{c}\right)}^{\mathrm{ref}}=I_{gp}^{c}\sin\left(\theta_{g}+\frac{2\pi }{3}\right)+I_{gq}^{c}\cos\left(\theta_{g}+\frac{2\pi }{3}\right),$$
where $I_{gp}^{a}$ and $I_{gq}^{a}$ represent the active and reactive current components imposed to accomplish the power references, respectively.

In order to introduce the power references in (Equation 13) and (Equation 16), two new variables are introduced, $m_{p}$ and $m_{q}$. $m_{p}$ is defined as the proportion of active power respect the apparent power and is computed as illustrated in Figure 4. Mathematically:(25)$$m_{p}=p_{g}^{\mathrm{ref}}/\left|{\vec{s_{g}}}^{\mathrm{ref}}\right|,$$
where $m_{q}$ represents the proportion of reactive power with respect to the apparent power
(26)$$m_{q}=q_{g}^{\mathrm{ref}}/\left|{\vec{s_{g}}}^{\mathrm{ref}}\right|.$$

The proportion of power indices has the following property
(27)$$\sqrt{{\left(m_{p}\right)}^{2}+{\left(m_{q}\right)}^{2}}=1;$$
similarly, the current amplitude can be found as:(28)$$\sqrt{{\left(I_{g,p}^{l}\right)}^{2}+{\left(I_{g,q}^{l}\right)}^{2}}=I^{l},$$
where $I^{l}$ is the current amplitude of phase *l*. Then, the active and reactive current amplitude references can be defined as:(29)$$I_{g,p}^{l}=I^{l}m_{p},I_{g,q}^{l}=I^{l}m_{q}.$$

Then, from (Equation 29):(30)$$I^{l}=I_{g,p}^{l}/m_{p},$$
leading to defining the reactive current as:(31)$$I_{g,q}^{l}=I^{l}m_{q}=I_{g,p}^{l}\left(m_{q}/m_{p}\right).$$

Therefore, the current reference can be written as:(32)$${\left(i_{g}^{a}\right)}^{\mathrm{ref}}=I_{g,p}^{a}\left[\sin\left(\theta_{g}\right)+\frac{m_{q}}{m_{p}}\cos\left(\theta_{g}\right)\right],$$
(33)$${\left(i_{g}^{b}\right)}^{\mathrm{ref}}=I_{g,p}^{b}\left[\sin\left(\theta_{g}-\frac{2\pi }{3}\right)+\frac{m_{q}}{m_{p}}\cos\left(\theta_{g}-\frac{2\pi }{3}\right)\right],$$
(34)$${\left(i_{g}^{c}\right)}^{\mathrm{ref}}=I_{g,p}^{c}\left[\sin\left(\theta_{g}+\frac{2\pi }{3}\right)+\frac{m_{q}}{m_{p}}\cos\left(\theta_{g}+\frac{2\pi }{3}\right)\right].$$

However, the current amplitudes depend upon the voltage amplitudes. Thus, unbalanced voltages lead to unbalanced current as explained before. The amplitudes are computed as follow:(35)$$I_{g,p}^{a}\left(k\right)=\sqrt{2}i_{g,p}^{\mathrm{ref}}{\left[V_{gmax}^{a}\left(k\right)\right]}^{2},$$
(36)$$I_{g,p}^{b}\left(k\right)=\sqrt{2}i_{g,p}^{\mathrm{ref}}{\left[V_{gmax}^{b}\left(k\right)\right]}^{2},$$
(37)$$I_{g,p}^{c}\left(k\right)=\sqrt{2}i_{g,p}^{\mathrm{ref}}{\left[V_{gmax}^{c}\left(k\right)\right]}^{2},$$
where $i_{g,p}^{\mathrm{ref}}$ comes from (Equation 10), which is given as a function of the required active power. Thereby, if a phase is overloaded and the corresponding voltage amplitude is reduced, the current drained from that phase is squared proportional to the voltage reduction. The current control reference is depicted in Figure 5.

### 3.5. Phase Locked Loop (PLL) for Unbalanced Grid Condition

Current control is based on resonant control, as detailed in the next section, considering fixed *N* samples per period. To achieve this goal, the first step is to consider the voltage positive sequence because it obtaining a balanced three-phase signal in phase with the original unbalance grid voltage signal. The positive sequence [21,22] can be obtained as:(38)$$\left(\begin{array}{c} {{\vec{v}}_{+}}^{a}\\ {{\vec{v}}_{+}}^{b}\\ {{\vec{v}}_{+}}^{c} \end{array}\right)=\frac{1}{3}\left(\begin{array}{ccc} 1 & \vec{\alpha } & {\vec{\alpha }}^{2}\\ {\vec{\alpha }}^{2} & 1 & \vec{\alpha }\\ \vec{\alpha } & {\vec{\alpha }}^{2} & 1 \end{array}\right)\left(\begin{array}{c} {{\vec{v}}_{g}}^{a}\\ {{\vec{v}}_{g}}^{b}\\ {{\vec{v}}_{g}}^{c} \end{array}\right),$$
where $\vec{\alpha }=e^{j2\pi /3}$, ${\vec{v}}_{g}$ is the grid voltage phasor, and ${\vec{v}}_{+}$ is the grid voltage’s positive sequence. $\vec{\alpha }$ represents a 120° phase shift; therefore, the following identity can be employed to be able to apply (Equation 38) in the time domain:(39)$$X\sin\left(\theta \left(k\right)+\beta \right)=X\sin\left(\theta \left(k\right)\right)\cos\left(\beta \right)+\sin\left(\beta \right)X\cos\left(\theta \left(k\right)\right),$$
where β represents the phase shift, which represents 120° or −120° for α and $\alpha^{2}$, respectively.

Then, the positive sequence of phase *a* is defined as:(40)$${{\vec{v}}_{+}}^{a}=\frac{1}{3}\left({{\vec{v}}_{g}}^{a}+\vec{\alpha }{{\vec{v}}_{g}}^{b}+{\vec{\alpha }}^{2}{{\vec{v}}_{g}}^{c}\right)$$
the phase shift of 120° for phase *b* in (Equation 40) given by the term $\vec{\alpha }{{\vec{v}}_{g}}^{b}$, which can be found in the time domain as:(41)$$v_{g}{{}^{b}}_{2\pi /3\;phase-shifted}\left(k\right)={v_{g}}^{b}\left(k\right)\cdot \cos\left(2\pi /3\right)+\sin\left(2\pi /3\right)\cdot v_{g}{{}^{b}}_{-\pi /2\;phase\;shifted}\left(k\right),$$
where $v_{g}{{}^{b}}_{-\pi /2\;phase\;shifted}\left(k\right)=-{v_{g}}^{b}\left(k-N/4\right)$, because the discrete sinusoidal always contains *N* samples per period and, therefore, a phase shift of 90° can be easily found. The same method can be applied for the −120° phase shift in the term ${\vec{\alpha }}^{2}{\vec{v}}_{g}^{c}$, leading to finding the positive sequence of phase *a* in the time domain as:(42)$${{\vec{v}}_{+}}^{a}=\frac{1}{3}\left\{\begin{array}{c} {{\vec{v}}_{g}}^{a}+{v_{g}}^{b}\left(k\right)\cdot \cos\left(2\pi /3\right)-\sin\left(2\pi /3\right)\cdot {v_{g}}^{b}\left(k-N/4\right)+\cdots \\ {v_{g}}^{c}\left(k\right)\cdot \cos\left(-2\pi /3\right)-\sin\left(-2\pi /3\right){v_{g}}^{c}\left(k-N/4\right) \end{array}\right\}.$$

In the same way, the positive sequence of phase *b* is:(43)$${{\vec{v}}_{+}}^{b}=\frac{1}{3}\left\{\begin{array}{c} {v_{g}}^{a}\left(k\right)\cdot \cos\left(-2\pi /3\right)-\sin\left(-2\pi /3\right){v_{g}}^{a}\left(k-N/4\right)+{v_{g}}^{b}\left(k\right)+\cdots \\ {v_{g}}^{c}\left(k\right)\cdot \cos\left(2\pi /3\right)-\sin\left(2\pi /3\right)\cdot {v_{g}}^{c}\left(k-N/4\right) \end{array}\right\}.$$

Phase *c* can be found as ${{\vec{v}}_{+}}^{c}=-{{\vec{v}}_{+}}^{a}-{{\vec{v}}_{+}}^{b}$.

Once the positive sequence is found, a traditional three-phase PLL can be used in order to obtain *N* samples per period, such as the one illustrated in [5].

The PLL in [5] enables a dot product between the internal variable ${u_{i}}^{abc}$ and the positive sequence ${v_{+}}^{abc}$, where ${u_{i}}^{abc}$ is defined as:(44)$$u_{i}\left(k\right)={\left[\begin{array}{ccc} \cos\mathrm{ine}\left(n_{0}\left(k\right)\right) & \cos\mathrm{ine}\left(n_{-2\pi /3}\left(k\right)\right) & \cos\mathrm{ine}\left(n_{+2\pi /3}\left(k\right)\right) \end{array}\right]}^{T},$$
where cosine() is an *N* times sampled cosine function; on the other hand, $n_{0}$, $n_{-2\pi /3}$ and $n_{2\pi /3}$ are integer numbers spaced by N/3 and pointing at a specific memory box of the cosine() table. The dot product between ${u_{i}}^{abc}$ and ${v_{+}}^{abc}$ results in
(45)$$u\left(k\right)=\langle u_{g}\left(t\right),u_{i}\left(k\right)\rangle =\left(3/2\right)U_{g}\sin\left(\theta_{g}\left(t\right)-2\pi n_{0}\left(k\right)/N\right),$$
where, if
(46)$$\theta_{i}\left(k\right)=2\pi n_{0}\left(k\right)/N=\theta_{g}\left(k\right)+2m\pi \Rightarrow u\left(k\right)=\langle {u_{g}}^{abc}\left(k\right),{u_{i}}^{abc}\left(k\right)\rangle =0$$
the internal $u_{i}$ variable is phase-locked to the positive sequence ${v_{+}}^{abc}$ when u(k)=0. Thus, the PI controller can be set in order to adjust the internal ${u_{i}}^{abc}$ variable to obtain zero in the dot product in (Equation 45). The entire unbalance PLL diagram block for an unbalanced power supply is represented in Figure 6.

### 3.6. Resonant Control

To track the current reference given in (Equation 32)–(Equation 34), a resonant control for every phase is implemented. The discrete transfer function of the proposed resonant control is written as follows:(47)$$h_{c}\left(z\right)=k_{c}\frac{\left(1-\vec{b}z^{-1}\right)\left(1-\bar{\vec{b}}z^{-1}\right)}{z^{-2}-2\cos\left(\omega_{0}T_{s}\right)z^{-1}+1},$$
where the poles are set on the unitary circle in order to reach zero steady-state error at the frequency $\omega_{0}=2\pi f_{g}$. On the other hand, the two zeros including on the numerator, (Equation 47), are included to ensure a stable closed loop current control.

As stated before, this study focused on variable frequency condition to change the parameters in (Equation 47) as a function of the grid frequency, because $\omega_{0}=2\pi f_{g}$. However, as we impose *N* samples per cycle, independent of the supply frequency, the sampling time is a function of the grid frequency as:(48)$$T_{s}\left(k\right)=1/\left(Nf_{g}\left(k\right)\right).$$

Then, the denominator second parameter can be transformed as:(49)$$2\cos\left(\omega_{0}T_{s}\right)=2\cos\left(2\pi f_{g}T_{s}\right)=2\cos\left(2\pi f_{g}\frac{1}{Nf_{g}}\right)=2\cos\left(2\pi /N\right)=a_{1},$$
where the last expression shows that the second parameter is a constant independent of the grid frequency, and the poles in (Equation 47) are constant and independent thereof. Thus, the controller transfer function is transformed to:(50)$$h_{c}\left(z\right)=k_{c}\frac{\left(1-\vec{b}z^{-1}\right)\left(1-\bar{\vec{b}}z^{-1}\right)}{z^{-2}-a_{1}z^{-1}+1}=\frac{u^{l}\left(z\right)}{e^{l}\left(z\right)},$$
where l={a,b,c}, and the controllers outputs are given by: (51)uabc(k)=a1uabc(k−1)−uabc(k−2)+kceabc(k)−2Reb→eabc(k−1)+|b→|2eabc(k−2),
where $u^{abc}$ is the resonant control output and:(52)$$e^{abc}\left(k\right)={\left(i_{g}^{abc}\right)}^{\mathrm{ref}}\left(k\right)-i_{g}^{abc}\left(k\right).$$

The most important part of (Equation 50) is that the poles do not need to be retuned as a function of the grid frequency. Additionally, it is not necessary to compute the cosines function (Equation 49) in every iteration, which demands a large computation effort.

In order to obtain a faster dynamic response under voltage supply variations (primarily sags and swells), feed-forward is implemented as shown in Figure 7c as
(53)$$m_{g}^{abc}\left(k\right)=u^{abc}\left(k\right)+v_{g}^{abc}\left(k\right)/v_{dc}\left(k\right).$$

The previous control is composed of the internal current loop, which is simple to implement, and the computational effort is notably reduced.

Thus, the power converter is totally defined, and the whole diagram is shown in Figure 7, including the power converter, the voltage amplitude analysis, the power control to regulate the DC link voltage, and the resonant control.

It is important to note that the proposed resonant control is slightly different from traditional methods, where the actual frequency is directly introduced in the control transfer function. This can be seen in [23,24], where the frequency is directly feed-forward as a variable parameter in the control loop. The main problem associated with this method of control is that the pole’s location needs to be changed as a function of the grid frequency. In other words, the denominator in (Equation 47) needs to be recomputed when the frequency changes (or in every iteration), i.e., $cos(\omega_{0}T_{s})$ needs to be recalculated, where trigonometric functions are one of the harder functions to be computed with real-time microprocessors. Thus, the proposed resonant control has the advantages of reducing computational time and/or permitting the application of the proposed control even in digital boards without trigonometric function capabilities, only requireing the PLL to ensure $T_{s}=1/\left(Nf_{g}\right)$.

## 4. Pole Locations

The minimum requirement for any control system is to accomplish a stable closed loop. The stability is a function of the power converter and controller parameters. To corroborate the stability issue on the designed control, the transfer function of the RL input LPF can be written as:(54)$$h_{RL}\left(z\right)=\frac{1}{R_{g}}z^{-1}\frac{1-a_{g}}{1-z^{-1}a_{g}},\;a_{g}=e^{-\frac{L_{g}T_{s}}{R_{g}}}.$$

The previous equation shows that the poles mapped by the RL input filter are a function of the sampling time, i.e., a function of the grid frequency. However, considering N=200 (200 samples per period) and frequency from 1 Hz to 100 Hz, the discrete pole associated with the RL input filter barely changes, and the pole is very close to one. This is because of the pole’s proximity to the real axis; in other words, the inductance $L_{g}$ impedance is much greater than the resistance $R_{g}$. Therefore, this pole can be considered constant and independent of the grid frequency, and for the parameter in Table 1, it does no change in a wide range of frequencies.

The open loop transfer function $l\left(z\right)=h_{RL}\left(z\right)h_{c}\left(z\right)$ is given by:(55)$$l\left(z\right)=\frac{z^{-1}}{R_{g}}\frac{1-a_{g}}{1-z^{-1}a_{g}}\frac{k_{c}\left(1-\vec{b}z^{-1}\right)\left(1-\bar{\vec{b}}z^{-1}\right)}{z^{-2}-a_{1}z^{-1}+1}.$$

A low-pass filter is often introduced to reduce the switching harmonics feedback to the controller given the controller frequency response. Thus, a first-order low-pass filter is included to reject high-frequency noise:(56)$$l_{f}\left(z\right)=l\left(z\right)\frac{z^{-1}\left(1-\alpha \right)}{1-\alpha z^{-1}}.$$
where α is related to the cut-off frequency, such that $\alpha =e^{\omega_{c}T_{s}}$, where $\omega_{c}$ is the cut-off frequency in radians per second.

The root locus and the sinusoidal response are shown in Figure 8, where the two poles are located on the unitary circle to achieve zero steady-state error at the given frequency. In addition, the two zeros are needed to accomplish a stable closed-loop system. The last remaining pole is related to the *LPF* transfer function.

### Other Unbalanced Current/Voltage Control Techniques and Comparison

The unbalance can be related to the current, when the load drains an unbalanced power from the grid, or in the grid supply, when the voltage supply voltage is unbalanced, mainly faulting in the line. Researchers [25] proposed a network reconfiguration to avoid the problems associated with current and voltage imbalances, which is very important in distribution lines because most of the house loads are single phase; therefore, to be realistic, we considered unbalanced behavior; thus, this paper proposes a selective bio-inspired metaheuristic selective bat algorithm together with EPRI-OpenDSS software to manage the grid.

On the other hand, some other techniques are based on power converters that help the system by controlling the voltage and current. Within this area, in [26,27], a unified power quality converter (UPQC) capable of managing imbalances in the grid is presented. In this case, the UPQC injects a controlled voltage through a transformer in series with the load by employing a simple hysteresis band controller; then, the load always receives a balanced voltage supply. The results show that the system requires power from the other lines to help the one that is under fault; therefore, in some aspects, this is similar to what this paper presents but considering a series injection platform.

In [28], there is a review of power converter control working in some typical unbalanced operating scenarios: unbalanced voltage and/or current. This paper describes the power issues related to unbalanced conditions, including some aspects such as PV power generation and balancing the DC components’ multilevel converter under unbalanced three-phase conditions.

In [29], power converters with fault-tolerant operation capability are presented; the main issue is to continue working even under unbalanced supply conditions. A cost function is defined to balance the DC capacitors and the current control. The results show the system can remain stable even under unbalanced conditions, where the main objective is to eliminate the ripple in the active or reactive power.

In [30,31,32,33], the authors propose different control methods to overcome the current control in distorted voltage supply. In all these cases, the power converter is controlled in order to obtain a constant drained or injected power, independent of the actual grid voltage amplitude (or imbalance). Therefore, the currents need to be properly defined (including harmonics) to make the active or reactive power be constant in this distorted condition. Researchers [30] exhibited three ways to control the systems under unbalanced conditions, based on conventional controllers, decoupled double synchronous reference frame control, and proportional-resonant controller, with the last two providing better performance. On the other hand, in [31,32], predictive control methods for the currents to keep power as constant as possible are presented. Finally, ref. [33] shows generalized power control in order to manage active and reactive power as desired, where this paper helps the readers to properly define the inner control loop reference, in order to obtain constant power. This paper can be extended to any current control, resonant, predictive, or other control already employed in these scenarios.

Despite all recent articles presenting ways for power converters to work in unbalanced grids, none of them tried to relax the overload phase in order to help the system restore its state. Although many authors have associated the imbalance with grid faults, most of them only tried to keep the power converter stable and able to work under these circumstances. Thus, one distinction and contribution of the present work is to control the power converter and to help the grid to overcome the fault or issue by relaxing the overload phase.

## 5. Results

To corroborate the system in Figure 7 regarding the response according to the mathematical development, the grid-side converter was subjected to several tests. The tests performed in this study were (*i*) changes in the frequency, (*ii*) unbalancing in the grid voltage, (*iii*) step reference change on the DC link voltage, and (*iv*) changes in the current phase. In addition, the experimental results show ramp and step changes in the voltage amplitude and frequency, including unbalancing of the grid voltage.

### 5.1. Simulation Results

The proposed resonant controller was tested under grid supply faults using PSim software, with the parameters listed in Table 1. Figure 9 shows the controller response under frequency variations as a ramp, as shown in Figure 9k. The tests were performed under balanced and unbalanced voltage supplies. The test results under the balanced condition are shown on the left side in Figure 9, while the those under the unbalanced condition are shown on the right side in Figure 9. For a balanced voltage source, the current shows the desired behavior, maintaining the DC voltage at the desired value, showing the same balanced behavior, as expected. However, when the same test was performed imposing an unbalanced power supply, the current associated with the phase with a higher rms voltage value was also bigger than the others, thus reducing the power consumption to the phases with voltage sag.

The rectifier current controller was tested under power factor reference changes, as shown in Figure 10. First, the test started with a 0.8 inductive power factor reference, then changed to 0.8 capacitive, finally finishing with unitary power factor operation. The whole test was performed in an unbalanced voltage environment, achieving the correct tracking of the reference.

Figure 11 highlights the current control under the current step-up test. It was verified from the results that the performance was acceptable even under an unbalanced voltage supply, showing good dynamic behavior.

As shown in Figure 12, a test was performed considering a 10 % of the fifth and seventh harmonics in the grid voltage; in addition, a nonlinear load with 7Ω after a three-phase diode rectifier was included to see the performance of the power converter under both grid and nonlinear load. As can be seen in Figure 12b, the grid current presents some harmonics, produced by the grid voltage (Figure 12a). The DC link voltage shows a low ripple due to the system unbalancing and also because of the harmonic, but those harmonics do not affect the load because the load voltage results are sinusoidal despite the nonlinear load imposed on it. Even when the grid-side rectifier was not designed for harmonics, Figure 12 shows that the system can handle this kind of disturbance on the grid and/or load side.

### 5.2. Experimental Results

Having verified the theoretical concept of the proposed controller through simulation tests, a proof-of-concept prototype system as shown in Figure 1 was built to confirm the practical feasibility of the strategy by means of experimental tests. A commercially available Texas Instruments TMS320F28335 DSP board, typical in power converter applications, was used to implement the control algorithm presented in Figure 7, where in order to keep *N* samples per period, Timer 0 was employed to trigger the control every $T_{s}=1/\left(Nf_{g}\right)$ seconds. On the other hand, to emulate the distorted power source, a California Instrument CSW 5550 programmable source was employed, where the frequency and amplitude can be set in a computer in order to be able to test the control under different circumstances, as illustrated later in this section. The tests, which considered unbalanced supply voltage within a wide frequency range as required to assess the behavior of the controller, are presented in Figure 13 and Figure 14.

The first test considered a DC voltage reference step-up under the presence of an unbalanced supply voltage. The system response is shown in Figure 13a. In this test, the supply voltage of phase *c* was reduced in comparison with phases *a* and *b*. Accordingly, the current drawn from phase *c* and $i_{g}^{c}$ had the lowest amplitude, as expected. At approximately 325 ms, the step-up change in the voltage reference was applied. After the change, the DC voltage tracked the desired reference of 700 V after approximately 350 ms, with a small overshoot of less than 10%, showing correct operation, as expected. A second test was performed in which the power factor reference was changed from 0.8 inductive to 0.8 capacitive under unbalanced conditions, as shown in Figure 13b. The unbalance was produced by a reduction in the phase *a* supply voltage, whereas the other phases were their nominal voltages. From the results, it can be observed that the power factor control can deal with severe unbalance conditions, achieving the desired displacement power factor reference.

The next tests considered simultaneous unbalance conditions and frequency variations in the power supply. The first case considered a step increase of 100% on the grid frequency together with a step reduction in the phase *a* voltage amplitude to 50% of its nominal val (Figure 14a). Even in real-world scenarios, frequency and amplitude changes are usually ramp-like; step changes are a more severe test condition. The obtained results show an accurate control response, where the phase *a* current, which has a reduced supply voltage, is lower than its phase *b* counterpart, as expected to help the grid supply operation. It is important to remark that although not presented in the figure due to limitations, the DC link voltage maintains its value by tracking its reference.

A more real-world-like scenario is presented in Figure 14b, where a voltage ramp decrease in the phase *a* supply voltage was applied. The voltage ramp was such that a reduction of 50% in the voltage amplitude was obtained at the end of 10 cycles. It can be appreciated that in this case, as in the one before in Figure 14a, the controller reduced the current associated with the phase with lower voltage amplitude, to relieve the affected supply phase.

Then, a ramp-like change in the supply voltage amplitude and frequency was applied. A ramp voltage amplitude reduction to 50% together with a ramp frequency increase to 200% of each corresponding value were applied during 10 cycles, as shown in Figure 14c. The results show correct operation as in the previous cases, particularly as the supply voltage is reduced the current is reduced too, and the controller maintains a correct regulation of the supply currents, following the supply voltage frequency and maintaining low distortion waveforms.

Finally, Figure 14d presents the response of the system to a ramp supply voltage amplitude increase from 50% to 100% of its nominal value, with a simultaneous frequency increase to 200% of its nominal value in 10 cycles. Initially, the currents are unbalanced, as the phase *a* supply voltage is reduced as compared with that of the other phases. However, when the phase *a* voltage reaches the nominal balanced condition, phase *a* is balanced as well, as expected. Additionally, the currents show the same frequency as the supply voltage, pointing to tight control of the input currents.

## 6. Conclusions

The proposed resonant-control-based strategy showed a suitable response in a distorted voltage supply environment, particularly under frequency changes and unbalanced supply conditions. The use of resonant control enables proper operation in a wide frequency range without needing to retune the parameters offline. This characteristic is a consequence of the fast response accomplished by the sampling time controllers. In addition, the proposed unbalanced mitigation strategy, which loads each phase depending on its voltage sag, allows prevention of the worsening of the sag conditions and supports the supply’s correct operation. On the other hand, the complete control algorithm is easy to implement and requires minor computational effort. Furthermore, for variable grid frequency, having a constant *N* samples per cycle makes it possible to have a fixed desired resolution of the sensed AC variables, independent of the grid frequency. Although it was shown that the current amplitude references of each phase can be adjusted depending on the voltage unbalance proportion, the current references can also be set as balanced. This may be desirable if the power rating of the converter can be deemed negligible with respect to the total power supplied by the grid, as the power consumption is marginal and thus has a minimal effect on the system. Finally, even though the proposed strategy is designed for a weak-grid source, it can also be implemented in unbalanced systems with and without frequency variations. The simulation and experimental results showed that acceptable dynamic and static behavior is achieved under these conditions.

## Figures and Tables

**Figure 1 sensors-23-04884-f001:**
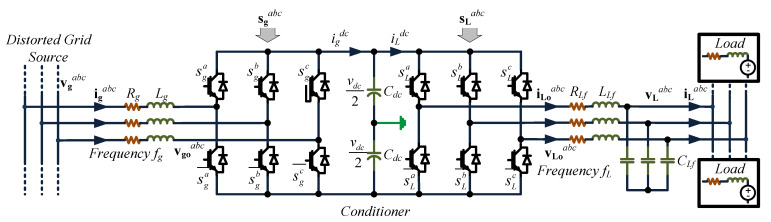
Power converter system.

**Figure 2 sensors-23-04884-f002:**
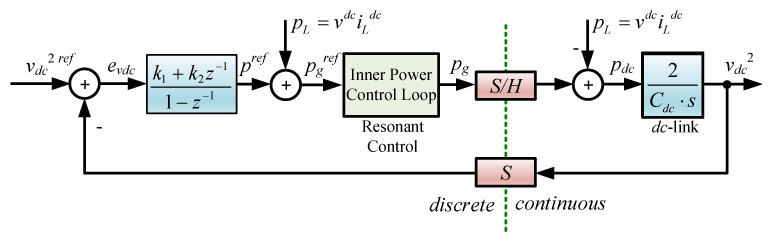
DC link voltage control.

**Figure 3 sensors-23-04884-f003:**
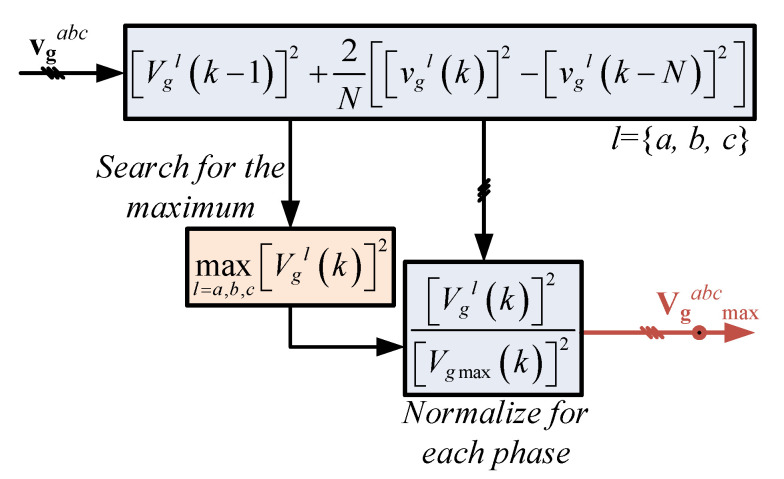
Voltage amplitude normalization.

**Figure 4 sensors-23-04884-f004:**
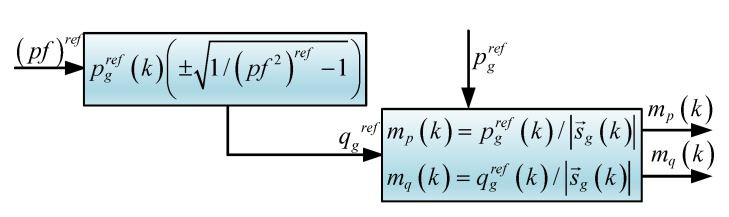
Power factor and coefficients of power.

**Figure 5 sensors-23-04884-f005:**
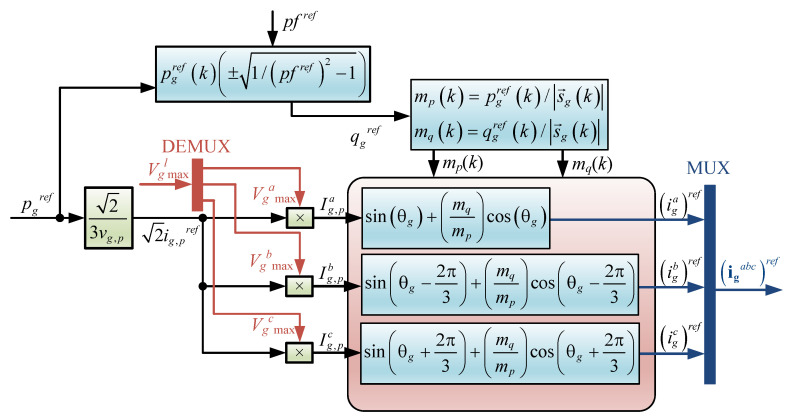
Current control reference.

**Figure 6 sensors-23-04884-f006:**
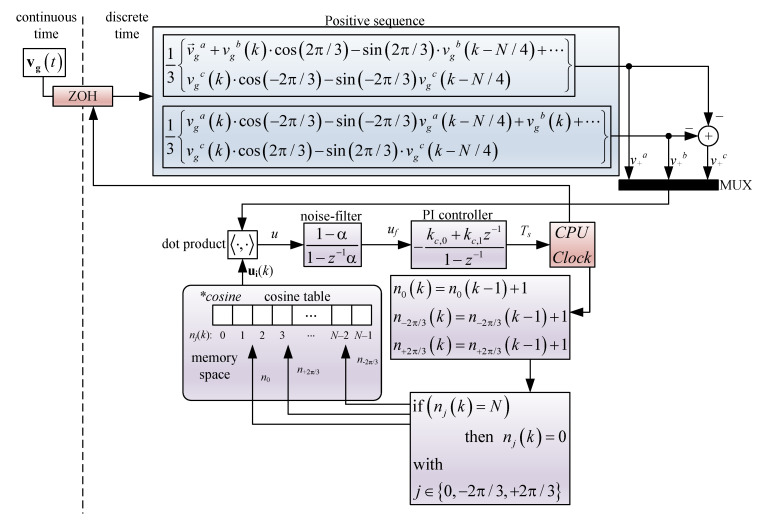
PLL algorithm for unbalanced power supply.

**Figure 7 sensors-23-04884-f007:**
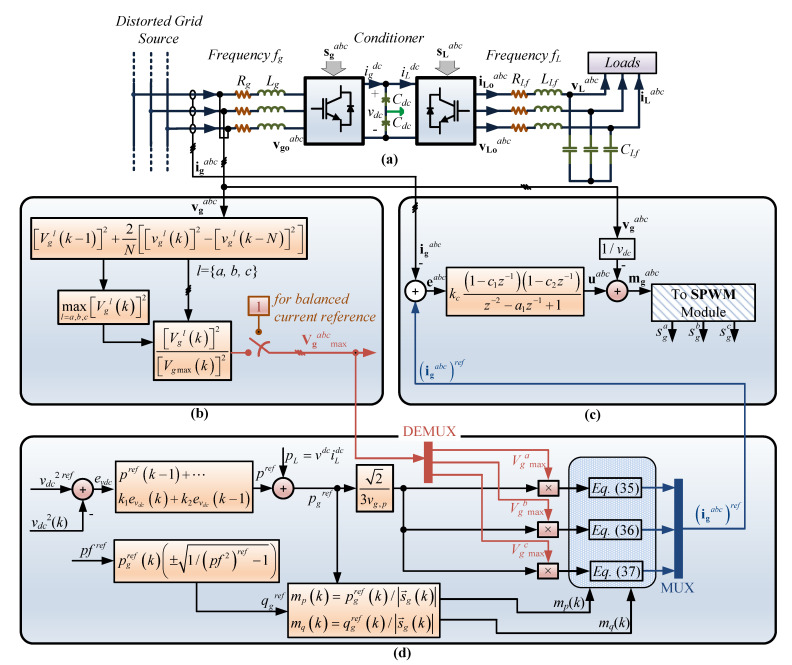
Resonant control diagram for unbalanced supply: (**a**) power converter; (**b**) voltage amplitude estimation; (**c**) DC voltage, power factor control, and current references; (**d**) resonant control algorithm.

**Figure 8 sensors-23-04884-f008:**
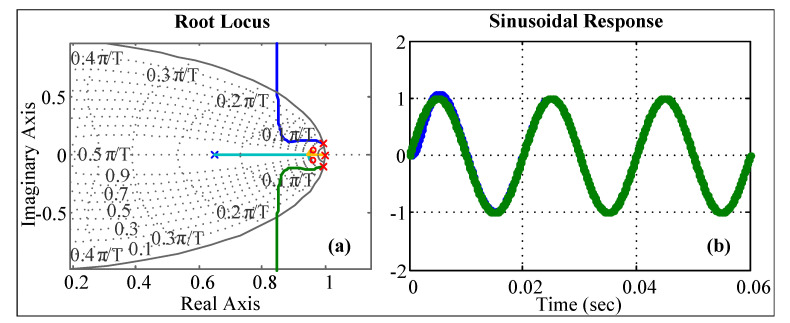
Root locus of the system including the controller. (**a**) Root locus and (**b**) sinusoidal response at 50 Hz.

**Figure 9 sensors-23-04884-f009:**
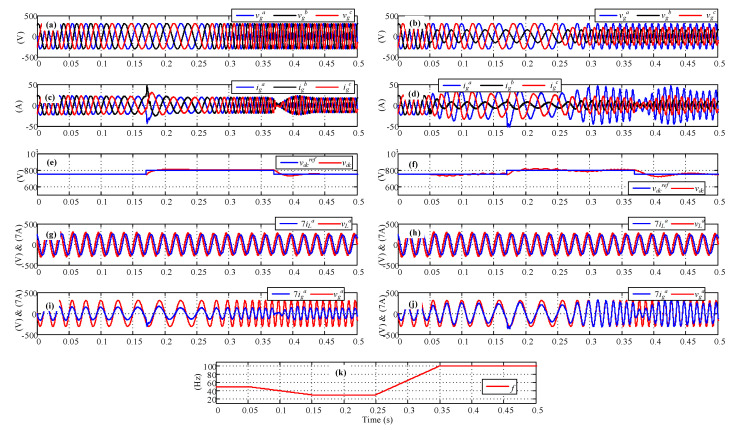
Simulation results. (*i*) Left-side balanced condition test and (*ii*) right-side unbalanced condition test. (**a**,**b**) Voltage supply, (**c**,**d**) three-phase grid currents, (**e**,**f**) DC link voltage response, (**g**,**h**) load voltage and current, (**i**,**j**) grid voltage and current, and (**k**) frequency.

**Figure 10 sensors-23-04884-f010:**
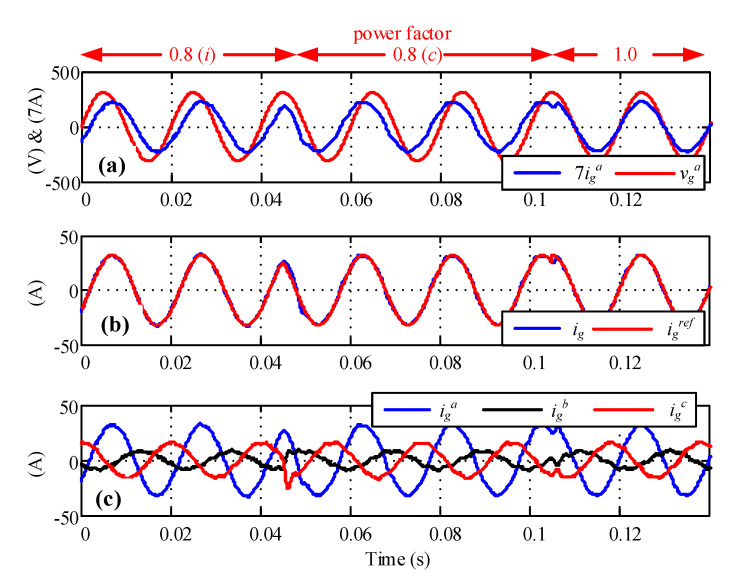
Power factor changes. (**a**) Voltage and current, (**b**) current and current reference, and (**c**) three-phase currents.

**Figure 11 sensors-23-04884-f011:**
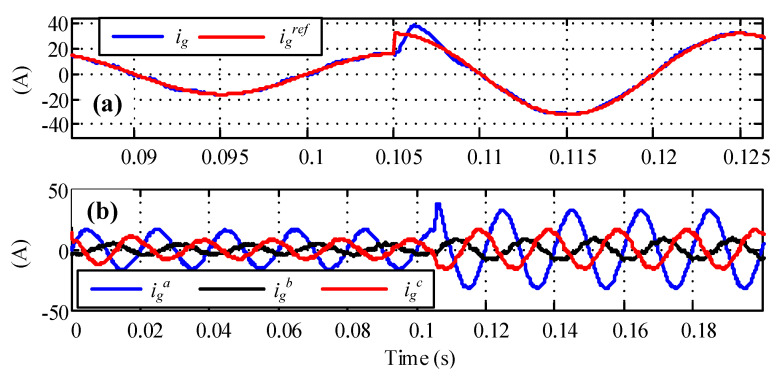
Current reference step. (**a**) Current response and (**b**) three-phase current response.

**Figure 12 sensors-23-04884-f012:**
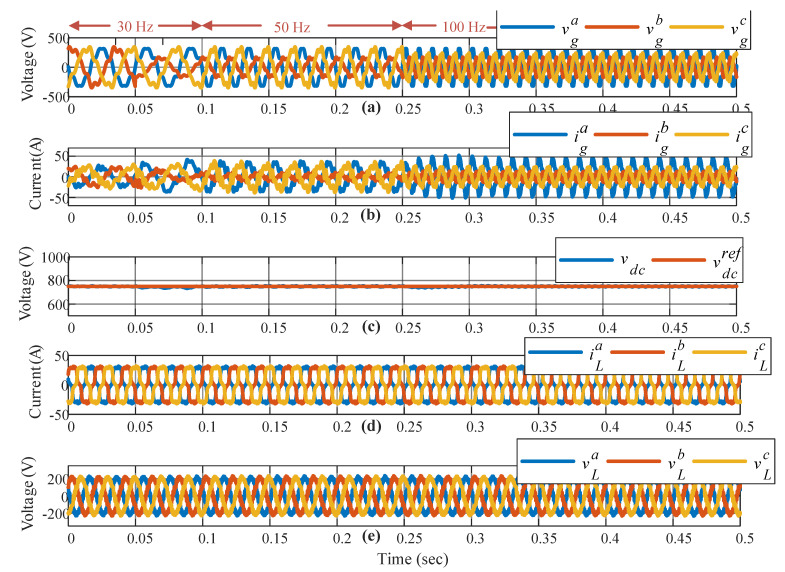
Response under harmonics on grid voltage and load current. (**a**) Three-phase grid voltage supply, (**b**) three-phase grid currents, (**c**) DC link voltage response, (**d**) three-phase load current, and (**e**) three-phase load voltage.

**Figure 13 sensors-23-04884-f013:**
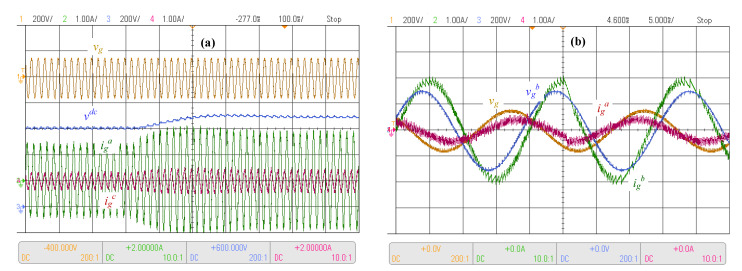
Experimental results with unbalanced supply and nominal frequency. (**a**) DC link voltage control and (**b**) power factor control step.

**Figure 14 sensors-23-04884-f014:**
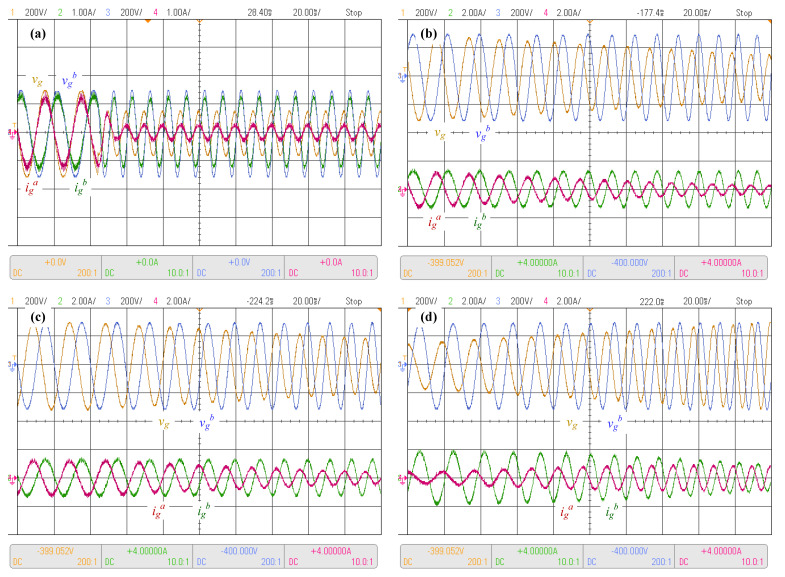
Experimental results with unbalanced supply and frequency changes. (**a**) Unbalanced and frequency 100% step, (**b**) source voltage ramp unbalanced, (**c**) voltage ramp-down and frequency ramp-up, and (**d**) voltage and frequency ramp-up.

**Table 1 sensors-23-04884-t001:** System parameters.

Parameter	Value	p.u. or Details
$v_{g}$	220 Vrms	1
$v_{dc}$	750 V	3.4
$Z_{L}$ (50 Hz)	8.8 Ω	1
$R_{L}$	7 Ω	0.80
$L_{L}$	17 mH	-
$X_{LL}$ (50 Hz)	5.3 Ω	0.61
$R_{g}$	0.1 Ω	0.011
$L_{g}$	7 mH	-
$X_{Lg}$ (50 Hz)	2.2 Ω	0.43
$C_{dc}$	4.7 mF	each capacitor
$X_{Cdc}$ (50 Hz)	1.3545 Ω	0.15
$R_{Lf}$	1 mΩ	0.11 m
$L_{Lf}$	2 mH	-
$X_{LLf}$ (50 Hz)	0.94 Ω	0.11
$C_{Lf}$	10 μF	-
$X_{CLf}$ (50 Hz)	318.38 Ω	36.172
$f_{g}$	50 Hz	1
$f_{L}$	50 Hz	1
$f_{sw}$	10 kHz	200
*N*	204	-
$T_{s}$	$1/(Nf_{g})=98$μs	for nominal value

## Data Availability

Not applicable.

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
