# Peer review of "Power Converter Resonant Control for an Unbalanced and Non-Constant Frequency Supply"

_sensors, 2023, doi:10.3390/s23104884_

Round 1

Reviewer 1 Report

Overall, this paper proposes a resonant control method to deal with voltage amplitude and frequency variations in the presence of a distorted power supply. This method is well described and validated in simulation and experiments. This is a novel method and the results are sound. I, therefore, recommend it be accepted by Sensors.

Author Response

Dear Reviewer,

We appreciate your words and feedback on our work, we really try to do our best.

Many thanks.

Reviewer 2 Report

Attached

Author Response

Dear reviewer,

Attached you can find our response to your comments. Thanks for the invested time and the review that surely improved our paper

Best regards

The authors

Reviewer 3 Report

Dear Authors,

The paper You submitted for review is interesting, however there are several issues that need to be addressed. First and foremost, the language must be improved: at certain points, the paper becomes hard to comprehend, and I had to rely on guesswork to find Your intent. The linguistic improvement might solve further issues which are listed below.

The paper proposes a resonant control system for the grid inverter operating with an unbalanced load. The controller follows varying grid frequency by adjusting its sampling rate and this way, there is no need to retune the controller on the fly. In addition, Authors propose a strategy to compensate asymmetrical loads between the phases by means of independent phase current control and energy transfer between the phases. Those two points are claimed to be the Authors own contribution.

The literature review is rather short, and out of 15 references, 3 are self-citations, which is quite excessive. It would be advised to extend the literature review.

The paper describes operation of the entire inverter, with extensive mathematical description of the controllers, however the experimental verification appears as if there were two independent inverters during testing.

Figure 1 should be moved closer to its initial reference in the text – it decreases readability

Figure 2 is quite complex, and takes a lot of effort to make out the Authors intentions. Some components are unclear. The potential reader has to go back and forth through the paper to understand the figure. It would be wiser to present a simpler form of the control loop, and then present it further in the paper in more detail.

One of the most confusing symbols introduced by the Authors is eVDC (e^Vdc) and so on – which implies exponential function – this is unacceptable.

Table 1 includes some of the system parameters – but those are mostly impedance. The inverter parameters are missing: switching frequency, DC link sizes, topologies, controller execution frequencies. In addition, the output impedance of the inverter seems rather high.

The adaptation of the resonant control execution wasn’t explained: how is the execution rate adjusted with the frequency – in general, the paper claims that the sampling rate is fixed to the grid frequency. How is this frequency measured, and how the sampling rate is changed on the fly?

How would the system behave in the presence of high frequency components or higher harmonics?

In the results section, the experimental results in figure 8 don’t show the DC link voltage. How is the DC link voltage affected by dynamic conditions?

It would improve the paper if the DC link voltage behaviour driven by the proposed algorithm was compared to any common control in an inverter, to compare the dynamics and pulsation against a typical reference point.

Author Response

Dear Reviewer,

We appreciate your valuable and detailed review. We have addressed your comments and suggestions as best as we could. We believe that the changes allowed us to greatly improve the document, which we appreciate. Please find the answers attached. The corresponding changes are highlighted in the new version of the paper.

Best regards,

The authors

Round 2

Reviewer 2 Report

The authors have considered and incorporated my suggestions. Paper can be accepted in the present form.